# Development of a Multiplex RT-PCR Assay for Simultaneous Detection of *Velarivirus arecae*, *Arepavirus arecae* and *Arepavirus arecamaculatum*

**DOI:** 10.3390/plants14233683

**Published:** 2025-12-03

**Authors:** Kexin Sun, Li Zhang, Zemu Li, Peng Zhao, Siyu Wan

**Affiliations:** 1School of Breeding and Multiplication (Sanya Institute of Breeding and Multiplication), Hainan University, Sanya 572025, China; sunkexin0407@163.com (K.S.); xiaozhang@hainanu.edu.cn (L.Z.); zemu@hainanu.edu.cn (Z.L.); peng@hainanu.edu.cn (P.Z.); 2School of Tropical Agriculture and Forestry, Hainan University, Haikou 570228, China

**Keywords:** multiplex RT-PCR, Areca palm velarivirus 1, Areca palm necrotic ringspot virus, Areca palm necrotic spindle-spot virus

## Abstract

Areca Palm Velarivirus 1 (*Velarivirus arecae*, APV1), Areca palm necrotic ringspot virus (*Arepavirus arecae*, ANRSV), and Areca palm necrotic spindle-spot virus (*Arepavirus arecamaculatum*, ANSSV) are major viral pathogens that cause significant economic losses in areca palm cultivation. Rapid and reliable detection methods are essential for the early diagnosis and management of these viruses in affected regions. Specific primers were designed based on the *Coat Protein* (CP) gene sequences of the three target viruses: APV1. A specific primer pair targeting the *coat protein* (CP) region was designed for APV1, while primer pairs for ANRSV and ANSSV were designed based on conserved sequences surrounding the Nla-VPg/Nla-Pro protease cleavage sites. A multiplex reverse transcription-polymerase chain reaction (multiplex RT-PCR) assay was subsequently developed to simultaneously amplify the target sequences. The multiplex RT-PCR detection system was optimized by adjusting critical parameters, including the annealing temperature, extension time, and number of cycles, to ensure high specificity and sensitivity. The optimized multiplex reverse transcription-polymerase chain reaction (multiplex RT-PCR) successfully yielded distinct amplification products for all three target viruses: 938 bp for APV1, 527 bp for ANRSV, and 250 bp for ANSSV. The size differences among the amplicons allowed them to be clearly distinguishable by 2% agarose gel electrophoresis. The optimal reaction conditions were determined to be an annealing temperature of 53.4 °C and 35 cycles. Applying the optimized multiplex RT-PCR method, we analyzed 414 field samples collected from Hainan province. APV1 was identified as the most prevalent virus, detected in 22.71% of the total samples. ANRSV and ANSSV were detected at significantly lower rates, in 3.86% and 0.2% of the samples, respectively. Virus detection in areca samples from Hainan Island revealed clear regional differences in disease incidence, with higher rates in the eastern and central regions—particularly Baoting, Lingshui, Wanning, and Qionghai—averaging 46.73%. Together, these results demonstrate that the developed multiplex RT-PCR is a sensitive and practical tool for the routine molecular diagnosis and epidemiological investigation of APV1, ANRSV, and ANSSV in areca palms.

## 1. Background

*Areca catechu* L., commonly known as areca palm, is widely cultivated in South and Southeast Asia [1]. In recent years, with the development of the areca planting industry and related processing sectors, the cultivation area of areca palm has expanded steadily. This expansion has been accompanied by an increase in viral disease incidence, posing serious threats to both yield and quality [2]. Currently, the main viral pathogens infecting areca palm include Areca Palm Velarivirus 1 (*Velarivirus arecae*, APV1), *Areca palm necrotic ringspot virus* (*Arepavirus arecae*, ANRSV), and *Areca palm necrotic spindle-spot virus* (*Arepavirus arecamaculatum*, ANSSV) [3,4]. Plants infected with Areca palm velarivirus 1 (APV1) typically exhibit yellowing that begins at the leaf tips and gradually spreads across the entire leaf, accompanied by a noticeable reduction in crown width [5]. As the disease progresses, plant growth is markedly inhibited, newly emerging leaves remain underdeveloped, and individual leaves become shorter. Mealybugs have been identified as the primary transmission vector of APV1 [6]. Currently, the detection of APV1 in areca palm mainly relies on enzyme-linked immunosorbent assay (ELISA) and reverse transcription polymerase chain reaction (RT-PCR) techniques. Using RT-PCR, the *coat protein* (CP) gene of APV1 is amplified, after which the amplified fragment is cloned into the prokaryotic expression vector pET28a-APV1-CP and transformed into Escherichia coli. Following induction and purification, the APV1-CP recombinant protein is successfully obtained, providing a foundation for further serological detection and functional studies of the virus [7]. ANSSV infection leads to chlorosis of the upper leaves and the formation of spindle-shaped necrotic lesions on the middle and lower leaves [8,9]. In contrast, plants infected with ANRSV exhibit necrotic ring spots on the leaves, sparse foliage, drooping basal leaves, and elongated internodes relative to healthy plants. The single-infection symptoms and co-infection symptoms of the above three viruses are shown in Figure 1. In previous studies, a reverse transcription polymerase chain reaction (RT-PCR) assay specific for the detection of ANRSV has been developed [10].

These viruses may occur either as single or mixed infections, leading to a decline in areca palm quality and commercial value, and consequently causing substantial economic losses. Several diagnostic techniques for virus detection have been developed to date [11,12,13]. Among them, loop-mediated isothermal amplification (LAMP), single-strand RT-PCR, uniplex real-time RT-PCR, and enzyme-linked immunosorbent assay (ELISA) have been applied for the detection of areca palm viruses [14,15]. However, these methods are designed to detect only a single virus at a time, which greatly reduces detection efficiency and limits the ability to provide timely protection for the areca palm industry. Moreover, symptoms caused by different viruses frequently overlap and are not always diagnostic, highlighting the urgent need to establish a rapid, sensitive, and specific detection method capable of simultaneously identifying APV1, ANRSV, and ANSSV in field samples. To date, multiplex RT-PCR has been widely utilized for detecting various viruses in crops such as wheat, maize, soybean, and rice [16,17,18,19]. By amplifying multiple nucleic acid fragments within a single reaction, multiplex RT-PCR enables the simultaneous, rapid, and sensitive detection of several areca palm viruses [20,21,22], thereby significantly reducing diagnostic costs and improving the efficiency of large-scale virus surveillance [23].

In this study, three specific primer pairs that can amplify DNA fragments of different sizes were designed according to genomic sequences of APV1, ANRSV and ANSSV. After optimizing RT-PCR conditions, an efficient multiplex RT-PCR assay was established and validated for the detection of APV1, ANRSV and ANSSV infecting areca plants.

## 2. Methods

### 2.1. Plant Materials

Fresh leaves were collected from areca palms grown in greenhouses at Hainan University (Haikou, China). These samples included both healthy leaves and leaves that tested positive for APV1, ANRSV, or ANSSV, allowing us to validate the specificity and sensitivity of the multiplex RT-PCR assay. Subsequently, field samples of Areca palms were collected from plantations across various cities and counties. A total of 23 samples were collected from each field, immediately frozen in liquid nitrogen, and transported to the laboratory for further analysis. To establish and optimize the multiplex RT-PCR assay, samples previously confirmed by RT-PCR and sequencing were utilized. These confirmation samples included those with single infections (APV1, ANRSV, and ANSSV), those with double and triple mixed infections of the three viruses, and virus-free control seedlings.

### 2.2. RNA Extraction and Reverse Transcription

Total RNA was isolated from the areca leaf samples as described before [24]. First-strand cDNA was synthesized from 5 µg total RNA using a RevertAi (#K1622) reverse transcription kit (Thermo Fisher Scientific, Shanghai, China) in a 20 µL reaction mixture with random primers, according to the manufacturer’s protocol.

### 2.3. Design of Virus-Specific Primers

The genomic sequences of APV1 (accession numbers: OM687513.1, MK956940.2, MW316024.1, MW316023.1, MW31602a2.1, MW316019.1, MW316013.1, NC_027121.1, and KR349464.1), ANRSV (accession numbers: MZ209276.1, MW282956.1, NC_055501.1, MH425894.1, MH425890.1, MH395393.1, MH395380.1, MH395376.1, and MH395371.1), and ANSSV (accession numbers: MH330686.1) were obtained from GenBank. Nucleotide sequence alignment was performed using ClustalW implemented in MEGA X [25,26], and conserved regions of each virus were identified. Virus-specific primers were designed based on the identified conserved regions of each virus using Primer Premier 5.0 software (Premier Bio-soft International, Palo Alto, CA, USA). To ensure broad detection coverage and sensitivity for genetically diverse isolates of Areca palm velarivirus 1 (APV1), specific primers were designed targeting the highly conserved coat protein (CP, ORF6) region. Multiple sequence alignment of 23 representative APV1 isolates, including KR349464 and MW316004–MW316025, revealed a high nucleotide identity of 98–99% within the CP region (Table 1), making it a suitable and stable genomic target. The designed primers were located within the most conserved regions, minimizing the risk of false negatives caused by natural sequence variability.

For ANRSV and ANSSV, both members of the newly classified genus Arepavirus in the family Potyviridae, primers were designed around the conserved sequences flanking the Nla-VPg/Nla-Pro protease cleavage sites. Despite their phylogenetic closeness, this genomic region exhibits significant amino acid divergence—KVLE/C in ANRSV and XILE/C in ANSSV—providing an ideal target for specific primer binding. Sequence alignments demonstrated high intra-species conservation and inter-species variation within these regions, ensuring reliable and specific amplification in multiplex RT-PCR assays.

### 2.4. Evaluation of Primer Specificity

To evaluate the specificity of each primer pair, uniplex RT-PCR assays were conducted using the Multi PCR Kit (Sangon Biotech, Shanghai, China). Each reaction was performed in a total volume of 50 μL containing 2.0 μL of each primer (2 μM), 2 μL of cDNA template, 25 μL of 2× SanTaq PCR Mix, and 21 μL of double-distilled water. The thermal cycling conditions were as follows: initial denaturation at 94 °C for 5 min; followed by 35 cycles of denaturation at 94 °C for 30 s, annealing at 54 °C for 60 s, and extension at 72 °C for 60 s; with a final extension at 72 °C for 8 min. PCR products were separated on a 2% agarose gel in 0.5× TBE buffer, stained with ethidium bromide (EB), and visualized under UV light. The TIANGEN D2000 DNA Marker (MD114-02; TIANGEN Biotech, Beijing, China) was used as the molecular size standard.

To further verify the specificity of each primer set, selected PCR products were purified and cloned into the pMD19-T vector (TaKaRa, Dalian, China) for sequencing. The obtained sequences were aligned with reference sequences in the GenBank database using DNAMAN 5.0 software (Lynnon Biosoft, San Ramon, CA, USA).

### 2.5. Optimization of Multiplex RT-PCR Assay

The multiplex RT-PCR assay was optimized by adjusting the annealing temperature and the number of amplification cycles. A range of annealing temperatures (53.4 °C, 54.3 °C, 55.2 °C, 56.3 °C, 57.3 °C, 58.2 °C, 58.9 °C, and 59.5 °C) was tested to determine the optimal condition for simultaneous amplification. Additionally, different cycle numbers (20, 25, 30, 35, and 40) were evaluated. The PCR products were analyzed by agarose gel electrophoresis as described above. The conditions yielding distinct, non-overlapping bands of expected sizes were selected for the final multiplex RT-PCR protocol.

### 2.6. Sensitivity of Multiplex PCR Assay

To compare the sensitivity of the multiplex RT-PCR assay with that of uniplex RT-PCR, 10-fold serial dilutions of cDNA derived from areca samples co-infected with APV1, ANRSV, and ANSSV were prepared and used as templates. The amplification reactions were performed under the optimized conditions. PCR products were analyzed by 2% agarose gel electrophoresis in 0.5× TBE buffer, as described above.

### 2.7. Survey of Areca Viruses by Multiplex RT-PCR Assay

A total of 414 areca palm leaf samples were collected from different regions of China and analyzed using the developed multiplex RT-PCR assay for the detection of APV1, ANRSV and ANSSV (Table 2).

## 3. Results

### 3.1. Specificity and Compatibility of Primer Pairs

In the uniplex RT-PCR assays, each primer pair specifically amplified a single fragment of the expected size corresponding to APV1, ANRSV, or ANSSV without any non-specific products (Figure 2A–C). Plasmids carrying the target viral sequences were used as DNA templates in these assays, ensuring precise validation of primer specificity. In the multiplex RT-PCR assay, all three target fragments were successfully amplified in a single reaction using the mixed primer set, and no non-specific amplification was observed (Figure 2D). To further verify the specificity and compatibility of the designed primer pairs, the amplified fragments were individually cloned and sequenced. Sequence alignment analysis revealed that the amplicons shared high nucleotide identity with the corresponding reference sequences of APV1, ANRSV, and ANSSV in the GenBank database. No amplification product was observed in the negative control, indicating the absence of non-specific amplification and confirming the reliability of the assay. These results confirmed that the primers were specific and compatible and could be reliably used for the simultaneous identification of the three areca-associated viruses in the developed multiplex RT-PCR system.

### 3.2. Optimization of Multiplex RT-PCR Conditions and Evaluation of Primer Concentrations

To improve amplification efficiency and balance band intensities among target fragments, primer concentrations were adjusted based on the brightness of specific bands observed after electrophoresis. The final optimized primer volumes per 25 μL reaction were 0.5 μL for both APV1-F and APV1-R, 0.4 μL for both ANRSV-F and ANRSV-R, and 1.0 μL for both ANSSV-F and ANSSV-R.

The multiplex RT-PCR conditions were further optimized by adjusting the annealing temperature and the number of amplification cycles. Eight annealing temperatures (53.4 °C, 54.3 °C, 55.2 °C, 56.3 °C, 57.3 °C, 58.2 °C, 58.9 °C, and 59.5 °C) were tested in gradient PCR assays. As shown in Figure 3, amplification performed at temperatures between 53.4 °C and 57.3 °C consistently yielded three distinct, virus-specific bands. In addition, although the optimized annealing temperature for the multiplex RT-PCR was determined to be 53.4 °C, detectable amplification was also observed at 57.3 °C. This phenomenon is common in PCR assays, as amplification success is not exclusively determined by the theoretical melting temperature of the primers. Primers designed in conserved regions with strong template affinity, together with the tolerance of commercial Taq polymerases to slight variations in annealing temperature, can allow successful amplification even under suboptimal conditions. Therefore, the ability to obtain products at 57.3 °C further supports the robustness of the primer design and the assay conditions. However, notable differences in band intensities were observed across the temperature range for individual targets, with 53.4 °C producing the most balanced and robust signal.

Additionally, five different cycle numbers (20, 25, 30, 35, and 40) were evaluated to determine the optimal number of amplification cycles (Figure 4). Amplified products from 35 and 40 cycles clearly exhibited the three expected bands without non-specific amplification, suggesting that 35 cycles provided a suitable balance between sensitivity and amplification efficiency.

The final reaction system for the multiplex RT-PCR was 25 μL, comprising 15 μL of Premix Taq (TaKaRa Taq Version 2.0 plus dye), virus-specific primers at the optimized concentrations (APV1-F/R: 0.5 μL each; ANRSV-F/R: 0.4 μL each; ANSSV-F/R: 1.0 μL each), 3 μL of cDNA, and ddH_2_O to bring the total volume to 25 μL. The optimized thermal cycling conditions were as follows: 94 °C for 5 min; followed by 35 cycles of 94 °C for 30 s, 53.4 °C for 60 s (annealing and extension), and a final extension at 72 °C for 10 min. A 6 μL aliquot of each amplified product was subjected to electrophoresis on a 2% agarose gel in 0.5× TBE buffer.

### 3.3. Sensitivity of the Multiplex RT-PCR Assay

To evaluate the detection sensitivity of the multiplex RT-PCR assay, cDNA derived from areca samples co-infected with APV1, ANRSV, and ANSSV was subjected to 10-fold serial dilutions (from 10^0^ to 10^−7^), and both uniplex and multiplex RT-PCR assays were performed. PCR products were analyzed by agarose gel electrophoresis as described above. The results showed that APV1 could be detected at a dilution of 10^−3^, while ANRSV and ANSSV were detectable up to 10^−2^ (Figure 5). These findings indicate that the developed multiplex RT-PCR assay has high sensitivity, comparable to that of the corresponding uniplex RT-PCR assays.

### 3.4. Application of Multiplex RT-PCR Assay in Survey of Areca Viruses

To investigate the epidemiological characteristics of APV1, ANRSV, and ANSSV in areca-growing areas of Hainan Province. A total of 414 areca palm leaf samples exhibiting typical viral disease symptoms were collected from major cultivation areas in Hainan Province (Figure 6). Laboratory testing using the established multiplex RT-PCR assay revealed the incidence of the three viruses, as well as the occurrence of mixed infections (Table 2).

Representative electrophoresis results are shown in Figure 7, clearly demonstrating distinct and specific amplification bands for each virus. The results showed that 98 samples (23.67%) were infected with at least one virus. Among them, APV1 was the most prevalent, detected in 94 samples (22.71%), followed by ANRSV in 16 samples (3.86%) and ANSSV in 1 sample (0.2%) (Figure 8). Further analysis revealed that 92 samples (22.22%) were infected with a single virus, while mixed infections involving two or three viruses were found in 6 samples (3.86%).

## 4. Discussion

Designing and selecting appropriate primer combinations is critical for establishing an efficient multiplex RT-PCR detection system [27]. In this study, virus-specific primers were designed based on highly conserved genomic regions of each virus. The final primer set successfully amplified distinguishable products of the expected sizes—938 bp for APV1, 527 bp for ANRSV, and 250 bp for ANSSV. The size differences of over 100 bp enabled clear separation of the three bands on a 2% agarose gel, with no visible non-specific amplification. A lower agarose concentration would reduce resolution and potentially hinder detection sensitivity; thus, a 2% agarose gel was used to ensure clear differentiation of target amplicons.

Given the potential inhibitory effect of multiple primer sets within a single reaction on amplification efficiency and specificity, PCR conditions were optimized to improve assay performance. Optimization of annealing temperature and cycling number revealed that the highest amplification efficiency for all three targets occurred at 53.4 °C, and no additional gain was observed beyond 35 cycles (Figure 3 and Figure 4). Accordingly, 53.4 °C and 35 cycles were adopted as the optimal conditions for the multiplex RT-PCR assay. Assay sensitivity is another key criterion for evaluating the performance of multiplex systems. Sensitivity testing showed that the detection limits of the multiplex RT-PCR assay were generally comparable to those of uniplex RT-PCR. Although there was a 10-fold decrease in detection sensitivity for APV1 detection in the multiplex setting, this modest decline is consistent with results reported in similar studies [28]. This discrepancy may be partly due to competition in the multiplex RT-PCR assay for key reactants, such as Taq polymerase and dNTPs [29].

Viral infection represents a major constraint in areca palm cultivation. Using the developed multiplex RT-PCR assay, a total of 414 areca leaf samples collected from different regions of Hainan were tested. The results indicated that 98 samples (23.67%) were infected with at least one of the three viruses, and 3.86% showed coinfection. In this study, one areca sample was found to be simultaneously infected with APV1, ANRSV, and ANSSV. Given that previous studies have suggested a very low likelihood of ANRSV and ANSSV co-infecting the same leaf tissue, this result appears to be unusual [30].

The corresponding PCR bands were relatively faint, indicating the possibility of a false positive, although low-level co-infection cannot be entirely ruled out. This observation highlights the need for further validation and offers a new perspective for investigating mixed infections among areca-associated viruses. Virus detection results from areca samples indicated significant regional variation in the incidence of areca yellowing disease across Hainan Island. The disease is currently spreading in major areca-producing areas such as Wanning, Lingshui, and Qionghai, with higher infection rates observed in the eastern and central regions. Baoting, Lingshui, Wanning, and Qionghai showed the highest infection levels, with an average incidence of 46.73%. The consistency of these results with those of other studies further supports the reliability of the established detection system [31]. Spatially, the severity of the disease decreases from east to west, with milder cases reported in western areas such as Danzhou and Baisha. This spatial pattern is supported by our survey data, as cities located in the eastern region showed higher virus positivity rates, whereas cities in the western region exhibited lower positivity rates. Therefore, the decline in positivity rates from east to west provides direct evidence for the observed reduction in disease severity across the province.

Previous studies have shown that APV1 was transmitted by both *Ferrisia virgata* and *Pseudococcus cryptus* and caused YLD symptoms in betel palm seedlings. The warmer and more humid climate in the eastern and central regions is more conducive to the reproduction of mealybug populations and the spread of the virus. Moreover, these regions have a longer history and larger scale of areca cultivation, which may further contribute to the higher disease incidence. Although areca cultivation has expanded in the western regions in recent years, the overall planting area remains smaller compared to the east and central areas, potentially reducing the risk of disease transmission. Notably, with the rapid expansion of areca cultivation in Hainan, yellowing disease and other viral infections are becoming increasingly severe, posing a significant threat to the industry. The spread of these diseases has emerged as a critical bottleneck restricting the sustainable development of the areca industry.

The high infection rate may be attributed to the widespread use of virus-infected seedlings for propagation and the lack of effective virus management strategies. To mitigate the spread and impact of areca-associated viruses, it is essential to adopt virus-free planting materials, cultivate resistant or tolerant varieties, and implement sensitive detection methods for routine surveillance. The multiplex RT-PCR assay developed in this study provides a rapid, reliable, and cost-effective tool for the simultaneous detection of APV1, ANRSV, and ANSSV and holds promise for future epidemiological investigations and disease control programs in areca cultivation.

Although this study successfully established a multiplex RT-PCR system for the simultaneous detection of three viruses, the increased number of primers in multiplex reactions may impose stress on the amplification system, potentially leading to reduced efficiency or nonspecific amplification. This limitation becomes more pronounced when expanding the assay to include additional viruses, posing challenges to both stability and sensitivity. Therefore, future studies could consider integrating fluorescent probe-based assays or digital PCR platforms to achieve higher throughput and specificity. Moreover, the multiplex RT-PCR system developed in this study targets only the three major viruses currently prevalent in Hainan—APV1, ANRSV, and ANSSV—and does not encompass other potential areca-associated viruses or emerging variants. Future work should focus on broadening the detection spectrum and incorporating high-throughput sequencing technologies to refine the pathogen profiling and management strategies for areca viral diseases.

## 5. Conclusions

In this study, a multiplex RT-PCR assay was successfully developed for the simultaneous detection of three major areca-associated viruses—APV1, ANRSV, and ANSSV. The optimized assay exhibited high specificity and sensitivity, producing distinct amplification bands for each virus. Field application revealed a high prevalence of both single and mixed infections in symptomatic areca samples, indicating the widespread distribution of these viruses. This multiplex RT-PCR method provides a rapid, accurate, and cost-effective tool for virus detection, offering strong support for disease monitoring and management in areca cultivation.

## Figures and Tables

**Figure 1 plants-14-03683-f001:**
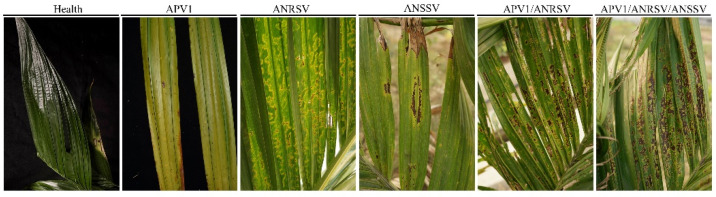
Diagram of leaf symptoms of areca palm infected by APV1, ANRSV, and ANSSV viruses individually and in combination.

**Figure 2 plants-14-03683-f002:**
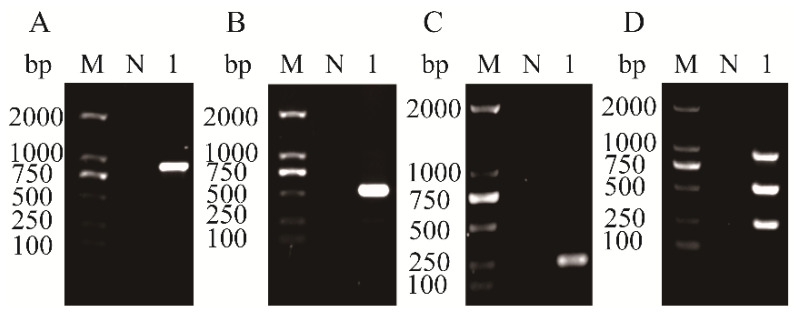
Determination of specificity and compatibility of three pairs of primers used for uniplex RT-PCR to detect APV1 (**A**), ANRSV (**B**), and ANSSV (**C**) and three multiplex RT-PCR assays of these three viruses (**D**). Lane M, 100 bp plus DNA ladder; Lane N, negative control.

**Figure 3 plants-14-03683-f003:**
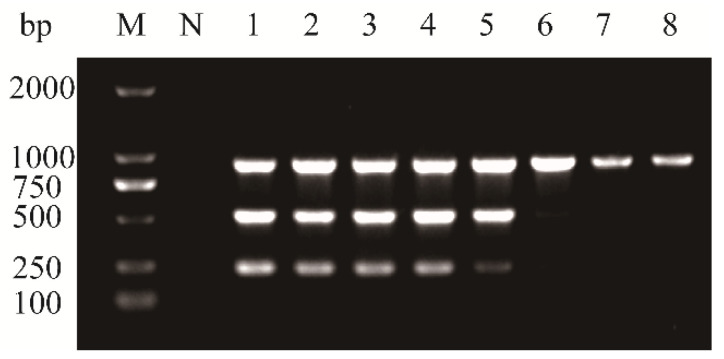
Optimization of annealing temperature for multiplex RT-PCR assay to detect APV1, ANRSV, and ANSSV. Lane M, 100 bp plus DNA ladder; Lane N, negative control; Lane 1–8, 53.4 °C; 54.3 °C; 55.2 °C; 56.3 °C; 57.3 °C; 58.2 °C; 58.9 °C; 59.5 °C.

**Figure 4 plants-14-03683-f004:**
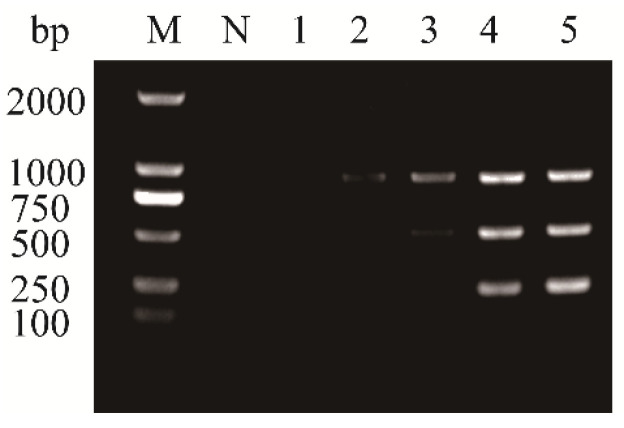
Optimization of amplification cycle number for multiplex RT-PCR assay to detect APV1, ANRSV, and ANSSV. Lane M, 100 bp plus DNA ladder; Lane N, negative control. Lanes 1–5, 20 cycles; 25 cycles; 30 cycles; 35 cycles; 40 cycles.

**Figure 5 plants-14-03683-f005:**
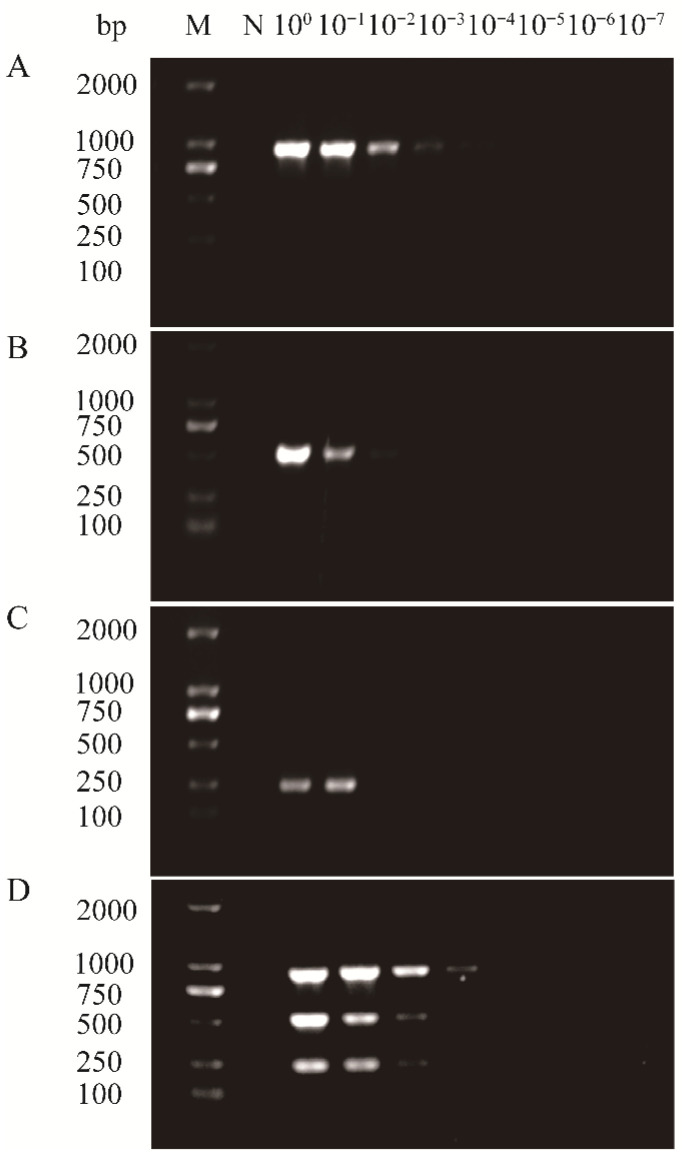
Comparison of the sensitivities of uniplex RT-PCR assays for the detection of APV1 (**A**), ANRSV (**B**), and ANSSV (**C**) and multiplex RT-PCR assay (**D**). Lane M, 100 bp plus DNA ladder; Lane N, negative control; Lanes100–10−7, ten-fold serial dilutions of cDNA from an areca sample co-infected with three viruses.

**Figure 6 plants-14-03683-f006:**
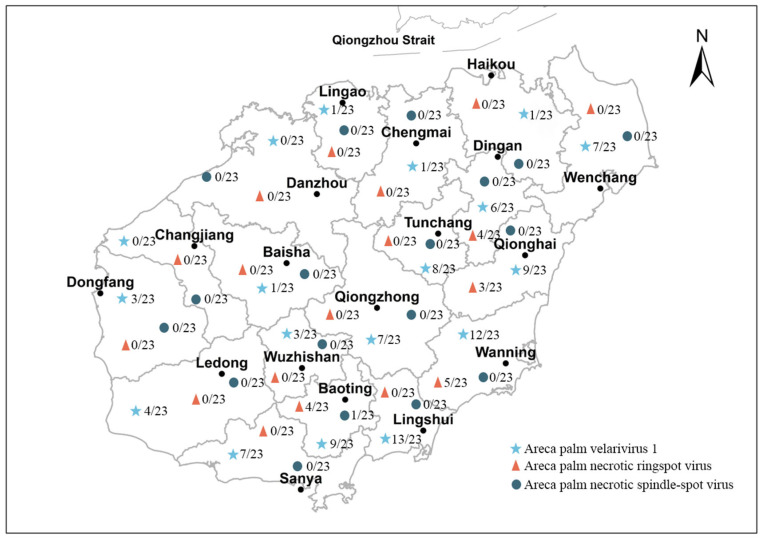
Sampling locations for the 414 areca leaf samples showing typical viral disease symptoms.

**Figure 7 plants-14-03683-f007:**
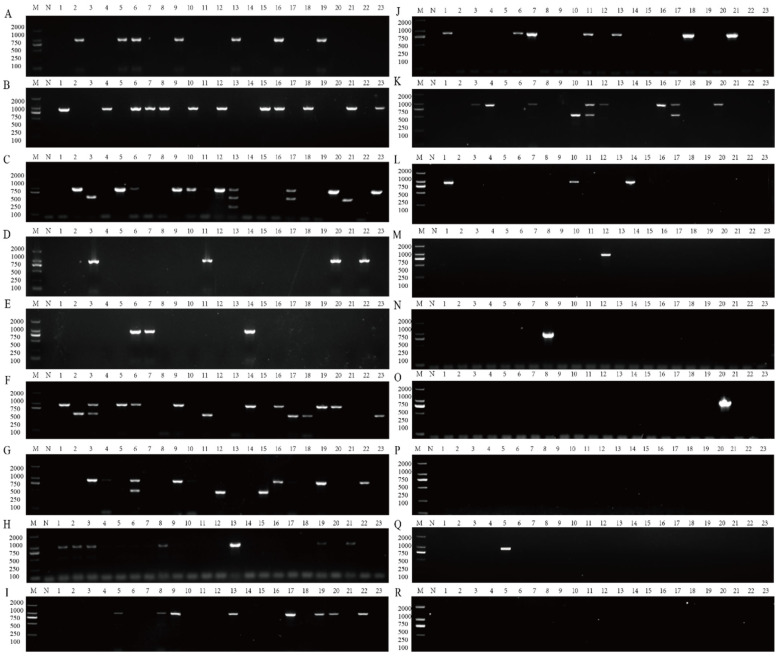
It illustrates the detection results of areca-associated viruses in leaf samples from 18 regions using the multiplex RT-PCR assay described in this study. A total of 23 leaf samples were tested from each region. M: DL2000 DNA marker; N: negative; lanes 1–23: individual areca leaf samples from each group; N: negative control; (**A**): Sanya; (**B**): Lingshui; (**C**): Baoting; (**D**): Ledong; (**E**): Dongfang; (**F**): Wanning; (**G**): Qionghai; (**H**): Qiongzhong; (**I**): Tunchang; (**J**): Wenchang; (**K**): Dingan; (**L**): Wuzhishan; (**M**): Haikou; (**N**): Chengmai; (**O**): Lingao; (**P**): Danzhou; (**Q**): Baisha; (**R**): Changjiang.

**Figure 8 plants-14-03683-f008:**
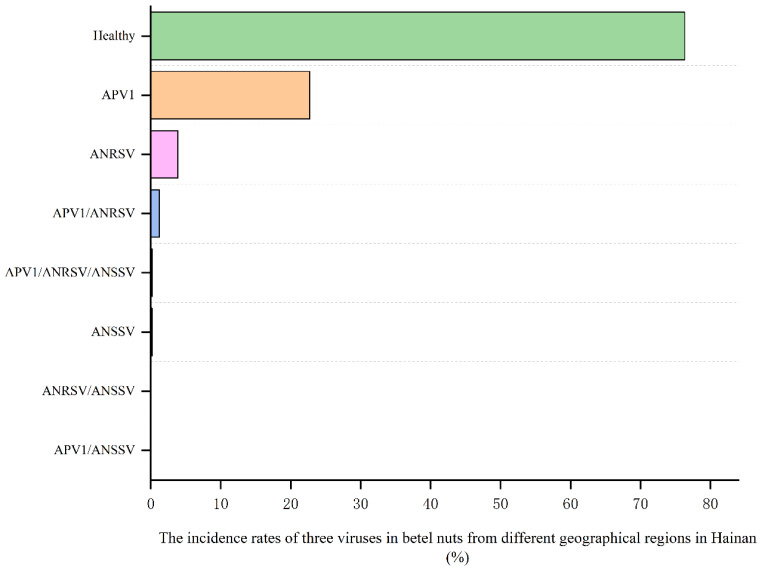
The incidence rates of three viruses in leave samples from different geographical regions in Hainan (%).

**Table 1 plants-14-03683-t001:** Primers for uniplex RT-PCR assay and multiplex RT-PCR assay.

Virus	Primer	Primer Sequence (5′–3′)	Tm (°C)	GC%	Position (nt)	Products (bp)	Target Gene
APV1	APV1-F	ATCGCTAAATATTATGGATAGACTT	52	28	12,768–12,792	938	CP
	APV1-R	TATTCAGAAGCATAAGATTGTGACA	54	31	13,681–13,705		
ANRSV	ANRSV-F	TCGCTGACATTGAAAAGG	52	44	5709–5726	527	Nla-VPg/Nla-Pro
	ANRSV-R	CTTAGCTGCTAGTCGCG	57	59	6219–6235		
ANSSV	ANSSV-F	CTCTAAAGGACAACAAACCA	50	50	8691–8708	250	Nla-VPg/Nla-Pro
	ANSSV-R	GAAAATTTGCGAAATCTGCATTGTC	54	42	8922–8940		

**Table 2 plants-14-03683-t002:** Detection of three viruses in areca plant leaves from different geographic region of Hainan using multiplex RT-PCR assay.

Location	No. of Samples	No. of Positive Samples and Positive Rate (%)		
		APV1	ANRSV	ANSSV
Sanya	23	7(30.00)	0(00.00)	0(00.00)
Lingshui	23	12(52.17)	0(0.00)	0(00.00)
Baoting	23	12(52.17)	4(17.39)	1(04.34)
Ledong	23	4(17.39)	0(00.00)	0(0.00)
Dongfang	23	3(13.04)	0(00.00)	0(00.00)
Wanning	23	9(39.13)	6(26.09)	0(00.00)
Qionghai	23	6(26.09)	3(13.00)	0(00.00)
Qiongzhong	23	7(30.00)	0(00.00)	0(00.00)
Tunchang	23	8(34.78)	0(00.00)	0(00.00)
Wenchang	23	7(30.00)	0(00.00)	0(00.00)
Dingan	23	8(34.78)	3(13.04)	0(00.00)
Wuzhishan	23	3(13.04)	0(00.00)	0(00.00)
Haikou	23	1(08.69)	0(00.00)	0(00.00)
Chengmai	23	1(04.34)	0(00.00)	0(00.00)
Lingao	23	1(04.34)	0(00.00)	0(00.00)
Danzhou	23	0(00.00)	0(00.00)	0(00.00)
Baisha	23	1(04.34)	0(00.00)	0(00.00)
Changjiang	23	0(00.00)	0(00.00)	0(00.00)

## Data Availability

Availability of data and materials The complete APV1, ANRSV and ANSSV genome sequences were deposited in GenBank with respective accession numbers MW316004–MW316024, MZ209276.1-MH395371.1, MH330686.1 and the datasets used and/or analyzed during the current study available from the corresponding author on reasonable request.

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
