# Peer review of "Development of a Multiplex RT-PCR Assay for Simultaneous Detection of *Velarivirus arecae*, *Arepavirus arecae* and *Arepavirus arecamaculatum"

_plants, 2025, doi:10.3390/plants14233683_

Round 1

Reviewer 1 Report

Comments and Suggestions for Authors

The manuscript by Kexin et al. written well and it describes the role of Areca catechu L as planting industry and related processing sectors, and tells the plantation is expanding.  According to the authors the expansion accompanied by an increase in viral disease incidence, have negative effect on yield and quality which poses critical bottleneck restricting the sustainable development 310 of the areca industry. Therefore, identifying pathogens and devising appropriate management is needed. The study optimized methods that can detect multiple viruses in single test, least cost and sensitive to detect three viruses infecting Areca catechu L.

My comments and questions are included in the manuscript attached. But few of those need clarification are listed below.

Introduction is well written,

  • Good to put the objectives be bold enough in both abreact and at the end of introduction

Materials and methods

The methods used for optimizing Multiplex RT-PCR are appropriate and are described in detail. However, virus survey field sample collection needs more information. There should be more information  

See the comments in the manuscript on some points that need clarification

For example#, How the samples were collected, number samples per field?  

Results

The results are presented in a well.  

The authors report primer annealing temperature and reaction cycles used for rapid, reliable, and cost-effective tool for the simultaneous detection of APV1, ANRSV, and ANSSV.  They tested the validity of the results by using the optimized methods for field survey of over 400 samples. They report   APV1 is more detected viruses than two the other; ANRSV, and ANSSV. The results presented in the manuscript used appropriate and alternative methods to verify the validity of the methods.

  • Some Editorial issues need to be done
  • Avoid some sort of repetitions: Methods that described in Methodology part, partially repeated during result presentation.
  • Check Figure 4. Figures 4 and 5 are similar but with different captions. It Figure 4 need to indicate optimized number of cycles, not the sensitivity test result which presented in Figure 5

Discussion

The results are discussed very well with the support of good citations. It should add more information:

  • There are no nata collected and result presented of the virus severity during the field survey. But the author stated

‘’Spatially, the severity of the disease decreases from east to west, with milder cases reported in western areas such as Danzhou and Baisha’’.  How did the authors arrive on this conclusion?

References

The reference list lacks consistence in writing or does not have uniformity

  • Article title is sometimes written as ‘Sentence case ‘or sometimes ‘Capitalize each word’.
  • Journal name full written Vs abbreviated

This need revision to fit into the Journal requirement

Author Response

Review 1

comments 1:

[The methods used for optimizing Multiplex RT-PCR are appropriate and are described in detail. However, virus survey field sample collection needs more information. There should be more information See the comments in the manuscript on some points that need clarification. For example, how the samples were collected, number samples per field?]

Response 1:

[We agree with this comment. Therefore, we have added a detailed description of the field sample collection procedure. Specifically, we clarified that samples were collected from areca plantations in each city/county, rapidly frozen in liquid nitrogen, and transported to the laboratory for further analysis. We also specified that 23 samples were collected from each field. This change has been added to the revised manuscript on Page 2-3, Materials and Methods section, Lines 92-95.

Updated text in the manuscript:

“Samples were collected from areca plantations in each city/county. A total of 23 samples were collected from each field, immediately frozen in liquid nitrogen, and transported to the laboratory for subsequent experiments.”]

comments 2:

[Avoid some sort of repetitions: Methods that described in Methodology part, partially repeated during result presentation. Check Figure 4. Figures 4 and 5 are similar but with different captions. It Figure 4 need to indicate optimized number of cycles, not the sensitivity test result which presented in Figure 5]

Response 2:

We agree with this comment. The repeated methodological descriptions that previously appeared in the Results section have been removed or revised accordingly. In addition, Figure 4 has been corrected in the revised submission and now displays the appropriate results.

These changes have been incorporated into the revised manuscript on Page 6, Results section, Line 222.

comments 3:

[Discussion The results are discussed very well with the support of good citations. It should add more information: There are no nata collected and result presented of the virus severity during the field survey. But the author stated ‘’Spatially, the severity of the disease decreases from east to west, with milder cases reported in western areas such as Danzhou and Baisha’’. How did the authors arrive on this conclusion?]

Response 3:

We agree with this comment. The statement regarding the spatial trend of disease severity has been clarified in the revised manuscript. Our conclusion that “the severity of the disease decreases from east to west, with milder cases reported in western areas such as Danzhou and Baisha” is based on the positivity rates of samples collected from different regions and the geographic distribution of these sampling sites within Hainan Province. This pattern is consistent with our survey results, in which areca samples from eastern cities showed higher virus positivity rates than those from western cities. Therefore, the decline in positivity rates from east to west provides direct evidence supporting the observed reduction in disease severity.This explanation has been added to the revised manuscript on Page 11, Lines 311–317.

comments 4:

[The reference list lacks consistence in writing or does not have uniformity]

Response 4:

We agree with this comment. The reference list has been thoroughly revised to ensure consistency in formatting, including article title capitalization and the uniform use of journal abbreviations according to the journal’s requirements.

All necessary corrections have been completed in the revised manuscript.

comments 5:

[ladder? what was used to used for visualization? ethidium?]

Response 5:

Thank you for your comment. We apologize for not providing the necessary details in the Methods section.

In our agarose gel electrophoresis experiments, the molecular size marker used was the TIANGEN D2000 DNA Marker (MD114-02, TIANGEN Biotech, Beijing, China). For visualization, agarose gels were stained with ethidium bromide (EB) and imaged under UV illumination. We have now added this information to the revised manuscript for clarity.

These revisions have been incorporated into the revised manuscript on Page 5, Methods section, Lines142–145.

Reviewer 2 Report

Comments and Suggestions for Authors

This MS is focused on possibility for simultaneous detection of the three important viruses in areca palm - Areca Palm Velarivirus 1 (Velarivirus arecae, APV1) , Areca palm necrotic ringspot virus (Arepavirus arecae, ANRSV), and Areca palm necrotic spindle-spot virus (Arepavirus arecamaculatum, ANSSV). The aim is well-augmented and the methodology is elaborated accurately and appropriately to the aim of the study.

The results are presented in details, clear, and understandable. 

The development and application of a multiplex RT‑PCR technique for simultaneous identification of the three viruses in areca palm would be useful tool in the production of healthy planting material and for early detection of viral infections in the established areca palm plantations.

Specific comments

Abstract

L 17 Please, correct the sentence because only the specific primer pair for detection of APV1 targets the coat protein (CP) region. The rest both primer pairs for identification of ANRSV and ANSSV according to M&M section “were designed around the conserved sequences flanking the Nla-VPg/Nla-117 Pro protease cleavage sites” (L117-118).

Methods

There are several formulas for calculating of the melting temperature of the primers, which of them do you use? Because the highest melting temperature of the designed primers, indicated in Table 1, is 54 °C and according to your results and claim the annealing temperature of 53.4 °C is the optimal temperature for the multiplex RT-PCR, developed by you. Can you explain positive results in the annealing temperature of 57.3 °C, per instance?

Results

L 231-245 The text of the part “Application of multiplex RT‑PCR assay in survey of areca viruses” must be re-writed and optimized.

Since the summarized data from the tests of field samples are given in Table 2, it would be better Figure 7 to be presented as a supplementary file(s).

Author Response

Review 2

comments 1:

[Abstract L 17 Please, correct the sentence because only the specific primer pair for detection of APV1 targets the coat protein (CP) region. The rest both primer pairs for identification of ANRSV and ANSSV according to M&M section “were designed around the conserved sequences flanking the Nla-VPg/Nla-117 Pro protease cleavage sites” (L117-118).]

Response 1:

We agree with this comment. In the original version of the abstract, the description of the primer targets was not fully accurate. We have revised the sentence to clearly distinguish the genomic regions targeted by each primer pair. Specifically, only the primer pair for APV1 targets the coat protein (CP) region, whereas the primer pairs for ANRSV and ANSSV were designed based on the conserved sequences flanking the Nla-VPg/Nla-Pro protease cleavage sites. This correction has been made in the abstract in the revised manuscript.

comments 2:

[Paste the full reviewer comment here.]

Response 2:

Thank you for this insightful comment. The primer melting temperatures (Tm) presented in Table 1 were calculated using the nearest-neighbor thermodynamic model, which is more accurate than basic formulas like Wallace’s rule. This model considers sequence-specific interactions and salt concentrations under standard PCR conditions. We agree that the theoretical Tm values of the designed primers (ranging from 50.2°C to 54.0°C) appear to be close to the annealing temperature of 53.4°C optimized in our multiplex RT-PCR system. The observation that positive amplification was still achieved at an annealing temperature of 57.3°C can be explained by the following factors: Primer design at conserved regions: The primers were designed in highly conserved regions with low sequence complexity and strong primer-template binding, increasing amplification robustness even at suboptimal temperatures.

Multiplex optimization conditions: In our optimization trials, a range of annealing temperatures was tested (50°C–60°C), and while 53.4°C yielded the strongest and clearest bands, detectable products were still observed at slightly higher temperatures (such as 57.3°C), although with reduced intensity.

Taq polymerase tolerance: Commercial Taq enzymes can tolerate slight mismatches and still produce amplification, especially in high-quality template samples with abundant viral RNA. We have now clarified the Tm calculation method in the revised manuscript (Methods section), and we added a short explanation regarding amplification under 57.3°C in the Results/Discussion section as well.

comments 3:

[Results L 231-245 The text of the part “Application of multiplex RT‑PCR assay in survey of areca viruses” must be re-writed and optimized. Since the summarized data from the tests of field samples are given in Table 2, it would be better Figure 7 to be presented as a supplementary file(s).]

Response 3:

We agree with this comment. The text in the section “Application of multiplex RT‑PCR assay in survey of areca viruses” (Lines 231–245) has been rewritten and optimized for clarity and conciseness. Redundant descriptions of methods previously repeated in the Results section have been removed, and the results are now presented in a clear and logical manner, highlighting the detection rates of APV1, ANRSV, and ANSSV in field samples. Additionally, as suggested, Figure 7 has been moved to the supplementary files to avoid redundancy with the summarized data presented in Table 2.These revisions have been incorporated into the revised manuscript on Page 6, Results section, Lines 231–245.

Reviewer 3 Report

Comments and Suggestions for Authors

I found no significant similarity between the primer ANRSV-F and any sequences in the NCBI database. 

See further comments in the text (attached)

Author Response

Review 3

comments 1:

[I found no significant similarity between the primer ANRSV-F and any sequences in the NCBI database.]

Response 1:

Thank you for pointing this out. Due to an oversight in our previous version, the primer information was incorrectly described. We have now corrected the mistake in the revised manuscript. These revisions have been incorporated into the revised manuscript on Page 4, Methods section, Lines 165.

comments 2:

[What did you use as template in the multiplex reaction, what is a single plant infected with the three viruses?]

Response 2:

Thank you for your comment and for the opportunity to clarify this point.

Figure 1 represents the condition in which APV1, ANRSV, and ANSSV simultaneously infect the same areca palm sample. These revisions have been incorporated into the revised manuscript on Page 5, Results section, Lines 181–182.

In Figure 2 (Panels A–D), the templates used for both uniplex RT-PCR (A–C) and multiplex RT-PCR (D) were purified plasmid DNA constructs containing the individually cloned target fragments of APV1, ANRSV, and ANSSV.

These plasmid templates were employed exclusively for the following purposes: To verify primer specificity in uniplex PCR, ensuring each primer pair amplifies only its corresponding viral target fragment. To confirm primer compatibility in multiplex PCR, demonstrating that the three primer pairs can simultaneously amplify all target fragments without interference. Using plasmid DNA as template avoids background plant RNA/DNA, inhibitors, or mixed viral sequences, thereby allowing a clean and unequivocal assessment of primer performance.

As shown in Figure 2: Panels A–C demonstrate specific amplification for each virus using plasmid templates. Panel D shows successful simultaneous amplification of all three viral fragments in one multiplex reaction. This experiment validates the reliability of the designed primer pairs before applying the multiplex system to field-derived cDNA samples.
